# High Expression of Casein Kinase 2 Alpha Is Responsible for Enhanced Phosphorylation of DNA Mismatch Repair Protein MLH1 and Increased Tumor Mutation Rates in Colorectal Cancer

**DOI:** 10.3390/cancers14061553

**Published:** 2022-03-18

**Authors:** Katharina Ulreich, May-Britt Firnau, Nina Tagscherer, Sandra Beyer, Anne Ackermann, Guido Plotz, Angela Brieger

**Affiliations:** Department of Medicine, Biomedical Research Laboratory, University Hospital Frankfurt, Theodor-Stern-Kai 7, 60590 Frankfurt, Germany; katharina.ulreich@kgu.de (K.U.); may-britt.firnau@kgu.de (M.-B.F.); nina.tagscherer@web.de (N.T.); s.beyer@med.uni-frankfurt.de (S.B.); anne.ackermann@kgu.de (A.A.); guido.plotz@kgu.de (G.P.)

**Keywords:** casein kinase 2, CK2α, MLH1, DNA mismatch repair, phosphorylation, colorectal cancer

## Abstract

**Simple Summary:**

Colorectal cancer (CRC) is associated with DNA mismatch repair (MMR) deficiency. The serine/threonine casein kinase 2 alpha (CK2α) is able to phosphorylate and inhibit MMR protein MLH1 in vitro. This study aimed to analyze the relevance of CK2α for MLH1 phosphorylation in vivo. Around 50% of CRCs were identified to express significantly increased nuclear/cytoplasmic CK2α. High nuclear/cytoplasmic CK2α level could be significantly correlated with reduced 5-year survival outcome of patients, increased MLH1 phosphorylation, and enriched somatic tumor mutation rates. Overall, our study demonstrated, in vivo, that enhanced CK2α leads to an increase of MLH1 phosphorylation, higher tumor mutation rates, and is an unfavorable prognosis for patients.

**Abstract:**

DNA mismatch repair (MMR) deficiency plays an essential role in the development of colorectal cancer (CRC). We recently demonstrated in vitro that the serine/threonine casein kinase 2 alpha (CK2α) causes phosphorylation of the MMR protein MLH1 at position serine 477, which significantly inhibits the MMR. In the present study, CK2α-dependent MLH1 phosphorylation was analyzed in vivo. Using a cohort of 165 patients, we identified 88 CRCs showing significantly increased nuclear/cytoplasmic CK2α expression, 28 tumors with high nuclear CK2α expression and 49 cases showing a general low CK2α expression. Patients with high nuclear/cytoplasmic CK2α expression demonstrated significantly reduced 5-year survival outcome. By immunoprecipitation and Western blot analysis, we showed that high nuclear/cytoplasmic CK2α expression significantly correlates with increased MLH1 phosphorylation and enriched somatic tumor mutation rates. The CK2α mRNA levels tended to be enhanced in high nuclear/cytoplasmic and high nuclear CK2α-expressing tumors. Furthermore, we identified various SNPs in the promotor region of CK2α, which might cause differential CK2α expression. In summary, we demonstrated that high nuclear/cytoplasmic CK2α expression in CRCs correlates with enhanced MLH1 phosphorylation in vivo and seems to be causative for increased mutation rates, presumably induced by reduced MMR. These observations could provide important new therapeutic targets.

## 1. Introduction

Casein kinase 2 (CK2) is a highly conserved serine/threonine protein kinase. In humans, there exist two catalytically active CK2 gene isoforms and one regulatory CK2 gene coding for the proteins CK2α, CK2α’, and CK2β. CK2α and CK2α’ are similar but show differences in their C-terminal sequences [1] and several studies have demonstrated that CK2α and CK2α’ also have different functions [2,3]. CK2 kinase isoforms are able to function as monomeric kinases, but in addition, CK2α and CK2α’ can either occur in a homodimer or heterodimer formation [4]. Furthermore, CK2 also exists as a tetrameric complex that consists of two CK2α and/or CK2α’ and two regulatory CK2β subunits. Within these tetrameric complexes, CK2β alters CK2 kinase substrate specificity [5]. The spectrum of CK2 activity includes more than 300 substrates. A large number of these proteins are involved in essential pathways of carcinogenesis [6]. Interestingly, CK2 has been shown to be abnormally elevated in many cancers, which might be causative for increased tumor aggressiveness through CK2-dependent phosphorylation of key proteins in signaling pathways [7]. Among those, proteins of the DNA mismatch repair (MMR) system have also been described. The phosphorylation of the MMR proteins mutS homologue 2 (MSH2) and mutS homologue 6 (MSH6) by CK2 result in an increased binding to mismatches [8], and we previously showed that the phosphorylation of the MMR protein mutL homologue 1 (MLH1) by CK2α led to a significant reduction of MMR in vitro [9].

MMR proteins are essential for the removal of DNA errors, which occur during replication. Loss of MMR plays an important role for the development of cancer, especially colorectal cancer (CRC) [10]. CRCs are classified using the Tumor, Node, and Metastasis (TNM) classification system developed by the Union for International Cancer Control (UICC). The prognosis of patients is determined by the TNM system and classified into one of four stages (stages I-IV) [11].

In around 3 to 5% of CRCs, a defective MMR system, caused by germline mutations in the MMR genes *MLH1*, *MSH2, MSH6*, *postmeiotic segregation increased 2* (*PMS2*), or in *Epithelial Cell Adhesion Molecule* (*EPCAM*) gene, is responsible for a hereditary disease called Lynch syndrome [12]. In a further 12% of sporadic CRC cases, MMR deficiency is caused by inactivation of *MLH1* via hypermethylation of the *MLH1* promoter [13]. In a rare number of CRC cases, a defective MMR is based on biallelic germline mutations in MMR genes, called constitutional MMR deficiency syndrome; more recently, the biallelic occurrence of two somatic MMR mutations were shown to explain some MMR-deficient CRCs [14].

In case of a defective MMR system, mismatched nucleotides highly accumulate in the genome, which finally result in microsatellite instability (MSI), and a high tumor mutational burden (TMB), as a hallmark of MMR-deficient tumors [15,16]. Loss of MMR proteins can lead to a concomitant reduction of other important proteins, making those tumors less responsive to current therapeutic regimens, e.g., such as treatment with FOLFOX [17]. It has been demonstrated that the estimation of TMB level is important for clinical application of *PD1/PD-L1*-argeting checkpoint inhibitors and metastatic MSI CRCs with high TMB, and more tumor infiltrating lymphocytes seem to benefit from therapy [18]. *PD1/PD-L1*-targeting checkpoint inhibitor therapy aims to block key regulators of the immune system and restore immune system function. The first drug approved in the U.S. was ipilimumab, a CTLA4 blocker, which was approved in 2011 [19].

However, CRC durable responses with PD1/PD-L1 inhibitors can be achieved in only approximately 40% of patients with MMR-deficient tumors; in patients with sporadic CRCs, it is unclear which markers can be used as a basis for potential therapy response. It should be noted that nearly 3% of CRCs, which are MMR proficient can be classified as microsatellite stable (MSS) and with high TMB. These MSS/TMB high cases might expand the population of CRCs who may benefit from immune checkpoint inhibitor-based therapeutic approaches [20,21].

The underlying mechanism for the generation of high TMB in MSS CRCs, however, is only partly understood. Fabrizio et al. as well as Gong et al. supported a DNA polymerase ε-mutated genotype within the MSS/TMB-high group that defects the MMR as well as the DNA proofreading pathway and contributes to an ultramutated but MSS phenotype in CRCs, without giving rise to the short tandem repeat signature observed through classic MSI testing [21]. The phosphorylation of MLH1 at position serine 477 (p-MLH1^S477^) by CK2α, which has been shown to significantly weaken and nearly switch off the MMR [9], might also cause a mutational signature without generation of a classical MSI phenotype.

In the current study, we used a cohort of 165 CRC patients to determine the expression level of CK2α. We compiled patients’ survival data in correlation with the intratumoral CK2α expression and analyzed in an exploratory cohort of patients if the level of CK2α expression is responsible for different amounts of p-MLH1^S477^. Furthermore, a panel of ~6800 genes was determined in order to figure out if CK2α overexpression correlates with enhanced tumor mutation rates, causes posttranslational phosphorylation, as well as inactivates MLH1 and causes significantly reduced activation of the MMR system. Lastly, the DNA region of the CK2α core promoter was screened for the presence of single nucleotide polymorphisms (SNPs) to determine whether somatic alteration in the promoter region of CK2α might play a role for its differential expression in CRC tissue.

## 2. Materials and Methods

### 2.1. Patients

Paraffin-embedded tissue (FFPE), well-characterized tumors, along with samples of the corresponding adjacent normal colonic mucosa, from 165 patients with CRC from our previously described cohort [22], were used in the present study. A number of 143 of these CRCs were MLH1-proficient, and twenty-two CRCs showed MLH1 deficiency. Characteristics of the individual tissue specimens are summarized in Appendix A. All patients underwent colonic resection with curative intent. Individuals with prior exposure to neoadjuvant chemotherapy were excluded from the study, in order to avoid interference from cytoreductive therapies that may conceivably alter tumor genetics. Resections were carried out between January 2011 and December 2016 at the University Hospital Frankfurt. The study was approved by the local ethics committee of the University Hospital Frankfurt, and all patients gave written informed consent.

### 2.2. Cells

HEK293 cells (ATCC^®^ CRL-1573^™^), purchased from the American Type Culture Collection (Rockville, MD, USA), and HEK293T cells, obtained from Dr. Kurt Ballmer (Paul Scherrer Institute, Villigen, Switzerland), were grown in DMEM with 10% FCS. As previously described, MLH1 is not expressed in HEK293T [23].

### 2.3. Antibodies

Anti-MLH1 (G168-728) and anti-PMS2 (A16-4) were purchased from Pharmingen (BD Biosciences, Heidelberg, Germany), anti-beta Actin (Clone AC-15) was from Sigma-Aldrich (Munich, Germany). Anti-phospho-AKT-substrate (23C8D2), used for the detection of phosphorylation of MLH1 at amino acid position serine 477 and hereinafter referred to as anti-p-MLH1, was obtained from Cell Signaling (New England Biolabs GmbH, Frankfurt, Germany). Anti-MLH1 (N-20), anti-CK2α (D-10), and anti-Lamin β (C-20) were from Santa Cruz (Santa Cruz Biotechnology, Heidelberg, Germany). Anti-MLH1 (ab74541) (Abcam, Cambridge UK) and anti-Adaptin γ (clone 88/Adaptin γ (RUO)) were from BD Biosciences (BD Biosciences, Heidelberg, Germany).

Anti-fluorescence-labeled anti-rabbit IRDye800CW and anti-fluorescence-labeled anti-mouse IRDye680LT were from LI-COR (LI-COR Biosciences GmbH, Bad Homburg, Germany).

### 2.4. Plasmids

The pZW6(CK2α) vector for the overexpression of CK2α constructs containing an HA-tag was a gift from David Litchfield (Addgene plasmid # 27086; http://n2t.net/addgene:27086; RRID:Addgene_27086, accessed on 1 June 2020) [24]. The pZW6 mock control plasmid was generated by cutting out the CK2α-HA cDNA sequence and by relegation of the remaining vector. The pcDNA3.1+/MLH1 and pcDNA3.1+/PMS2 expression plasmids have been described earlier [25], the pcDNA3.1+/MLH1^S477A^ variant has been previously described [9]. The pEGFP_C1 plasmid (negative control plasmid for transfection control) was purchased from Clontech Laboratories.

### 2.5. CK2α Promoter Amplification

The determination of potential CK2α promoter DNA variants in the exploratory panel of CRCs with high nuclear/cytoplasmic, high nuclear, and low nuclear/cytoplasmic CK2α expression was carried out by PCR amplification of ten overlapping fragments of the CK2α gene promoter region -1850 to +364 (NC_000020.11:472498-543790; [26]) using genomic DNA isolated from FFPE tissue. The following pairs of primers were used (Table 1):

### 2.6. Transient Transfection

HEK293T at 50–70% confluence were transiently co-transfected with pZW6(CK2α) (or empty pZW6 (mock control)), pcDNA3.1+/MLH1, and pcDNA3.1+/PMS2; with pcDNA3.1+/MLH1 and pcDNA3.1+/PMS2; with pcDNA3.1+/MLH1^S477A^ and pcDNA3.1+/PMS2; single transiently transfected with pEGFP_C1 (as negative control), using 20 µL/mL of the cationic polymer polyethylenimine (Polysciences, Warrington, PA, USA; stock solution 1 mg/mL). At 48 h post-transfection, cells were harvested and protein extracts or immunoprecipitated proteins were analyzed by Western blotting. All experiments were performed at least three times.

### 2.7. Protein Extraction

Extraction of total protein from FFPE tissue sections was performed using a Qproteome FFPE tissue kit (Qiagen, Germany). In brief, up to three serials of 5 µm sections were cut from paraffin-embedded, invasively growing colorectal specimens or corresponding normal adjacent mucosa, respectively. Sections were deparaffinized twice with xylene (100%) and rehydrated twice in three graded alcohol baths (100%, 96%, and 70%).

Protein extraction was performed by using 100 µL Extraction Buffer EXB Plus (provided in the Qproteome FFPE Kit (Qiagen cat. No. 37623) supplemented with 100 µM DTT equivalent to the manufacturer‘s instructions for 2-Mercaptoethanol. This was then followed by heating at 100 °C for 20 min and at 80 °C for 2 h with gentile agitation.

Whole-cell extract of HEK293 or transfected HEK293T cells were isolated by resuspending the cells directly in medium and centrifuging then at 3000 g for 3 min at 4 °C. The supernatant was discarded, the cells were resuspended in 1 mL PBS and centrifuged again. After discarding the supernatant, lysis reagent (CelLytic^™^ M Lysispuffer Sigma-Aldrich (St. Louis, Missouri, USA) combined with cOmplete^™^ Protease Inhibitor Cocktail Roche (Basel, CHE)) was added, and cells were lysed by ultrasound 4–6 times for 10 s, and centrifuged at 12,000 *g* for 10 min. The supernatant was stored at −20 °C until further use.

Separation of proteins into nuclear and cytoplasmic fractions of transfected HEK293T cells was carried out as described earlier [27]. In brief, transiently, pcDNA3.1+/MLH1 and pcDNA3.1+/PMS2, or pcDNA3.1+/MLH1^S477A^ and pcDNA3.1+/PMS2 co-transfected HEK293T cells were harvested 48 h after transfection by centrifugation, cell pellets were washed twice in PBS, and diluted in 250 µL hypotonic buffer (20 mM Tris-HCl pH 7.4, 10 mM NaCl, 3 mM MgCl2, and 0.5 mM DTT). After incubation on ice, 5% of NP-40 (10%) was added, samples were vortexed, and then centrifuged. Supernatants containing cytoplasmic proteins were frozen and residual pellets were resuspended in cell extraction buffer (10 mM Tris/HCl pH 7.4, 100 mM NaCl, 1 mM EDTA, 1 mM EGTA, 1 mM NaF, 20 mM Na4P2O7, 2 mM Na3VO4, 1% Triton X-100, 10% glycerol, 0.1% SDS, 0.5% Na-depoxycholate, 1 mM PMSF, and 5% protease inhibitor cocktail (Sigma Aldrich, Munich, Germany)). Samples were incubated on ice for 30 min, sonicated, and centrifuged. Dissolved nuclear protein fractions were frozen.

### 2.8. Western Blotting

Proteins were separated on 10% polyacrylamide gels, followed by Western blotting on nitrocellulose membranes and antibody detection using standard procedures. Fluorescent-labeled secondary antibodies (anti-mouse 680 LT from LiCor Bioscience, anti-mouse 800 LT from LiCor Bioscience, anti-rabbit 680 LT from LiCor Bioscience) were used to detect signals in a FLA-900 scanner (Fujifilm, Tokyo, Japan). If indicated, the band intensity of the protein expression was quantified using the Multi Gauge V3.2 program (Fujifilm, Tokyo, Japan).

The amount of p-MLH1^S477^ was detected after immunoprecipitation and quantified in correlation to total MLH1 levels using Multi Gauge V3.2. p-MLH1^S477^ levels were calculated by setting the expression of total MLH1 to 100% and putting the amount of p-MLH1^S477^ in relation to it.

All experiments were performed at least three times or as indicated. All the whole western blot figures can be found in the Appendix A.

### 2.9. Immunoprecipitation

Immunoprecipitations were carried out using 2000 µg of whole protein extract from FFPE tissue or HEK293 cells in a total volume of 1000 µL precipitation buffer (250 mM HEPES-KOH (pH 7.6), 100 mM NaCl, 100 mM EDTA, 0.2 mM PMSF, 200 mM DTT, 1% Triton X-100) with 4 µg of anti-MLH1 (G168-728). After agitated incubation at 4 °C for 1 h, 25–50 µL protein G sepharose (Santa Cruz Biotechnology, Heidelberg, Germany) were added and incubation continued for 4 h/overnight with gentile agitation. Precipitates were extensively washed in cold precipitation buffer using SigmaPrep^TM^ spin columns (Sigma, Munich, Germany). The sepharose was boiled in Laemmli sample buffer (Sigma Aldrich, Germany) for 5 min and proteins were separated on 10% polyacrylamide gels, followed by Western blotting on nitrocellulose membranes and antibody detection using standard procedures.

### 2.10. Immunohistochemical Staining

CK2α expression was analyzed by immunohistochemical staining using paraffin embedded, invasively growing MLH1-deficient and MLH1-proficient colorectal tumor tissue and corresponding surrounding normal mucosa, according to standard procedures. In brief, 2 µm sections of representative samples were cut from paraffin-embedded sample blocks and collected onto X-tra^®^ microscope slides and oven-dried overnight at 37 °C. Sections were deparaffinized twice for 5 min in 100% xylene and then rehydrated in a descending five-member alcohol series (100%, two times 90%, and two times 70%) for 2 min each. To remove the protein cross-links formed by formalin, antigen retrieval by heating was performed using 1 mM ETDA buffer (pH8) at 100 °C for 15 min. After cooling the sections under running tap water, the endogenous peroxidase activity was blocked using 3% H_2_O_2_ for 10 min at room temperature. Subsequently, the sections were washed in phosphate-buffered saline (PBS, Gibco, NY, USA) for 5 min each. This step was also repeated after all subsequent incubation steps. This was followed by incubation with primary antibody for 30 min at room temperature. Primary CK2α antibody (D-10: sc-365762, Santa Cruz Biotechnology, Heidelberg, Germany, 1:5000 dilution) was diluted in PBS containing 1% BSA.

Afterwards the sections were incubated with a horseradish peroxidase (HRP)-coupled secondary antibody (EnVision System mouse, K4000, Aligent, CA, USA) for 30 min at room temperature and then stained with chromogen 3,3-diaminobenzidine (DAB). The staining with DAB was carried out for 10 min in the dark, diluted to 1 drop of DAB chromogen per milliliter of DAB substrate buffer (K3467, Aligent, CA, USA).

Sections were counterstained using Gill’s hematoxylin solution. The sections were covered in Aquatex (Sigma Aldrich, Darmstadt, Germany).

Negative controls were processed in parallel to exclude non-specific staining.

### 2.11. Image Processing

Representative images of the immunohistochemical stainings were obtained using a digital slide scanner (3DHISTECH, Sysmex, Budapest, Hungary). Subsequently, separate image sections of the tumor and tumor-surrounding normal colorectal tissue were created at a 10-fold magnification using the Case Viewer program (3DHISTECH, Sysmex, Budapest, Hungary). The semi-quantitative analysis of the images was carried out using ImageJ (NIH) as previously described [22]. In brief, the images were converted into 8-bit greyscale images and inverted. Each pixel of the image was assigned an intensity value between 0 and 255. To exclude non-specific low background pixels and intensities from the analysis, a lower threshold of 50 was set. Finally, the mean intensity values for all pixels of an image were generated by measurements in ImageJ.

### 2.12. DNA Extraction

Three micrometer sections of representative samples were cut from paraffin-embedded, invasively growing colorectal carcinoma specimens. Sections were oven-dried at 70 °C for 30 min. This was followed by deparaffinization in 100% xylene for 20 min and rehydration in 100% isopropanol for 15 min. The tumor tissue was separated from the surrounding tissue by microdissection and DNA was isolated using the QIAamp DNA Micro Kit (Qiagen) according to the manufacturer’s instructions. Briefly, sample lysis was performed by digestion with Proteinkinase K overnight until the sample was completely lysed. Followed by a heating step at 90 °C for 30 min, loading and washing steps were performed using a Qiamp Spin Column (Qiagen, Köln, Germany). Purified DNA was eluted from the column in 20 µL RNAse-free water and concentration was measured using a NanoDrop (PEQLAB Biotechnologie GmbH, Erlangen, Germany).

### 2.13. Quantitative Reverse Transcription PCR

Total RNA was extracted from deparaffinized FFPE tissue using the PureLink™ FFPE Isolation Kit (Invitrogen, Carlsbad, CA, USA) or RNeasy FFPE Kit (Qiagen, Hilden, Germany) according to the manufacturers’ protocols. First-strand cDNA was prepared from 1 µg RNA with 50 ng/µL random hexamer primers using SuperScript™ III First Strand Synthesis SuperMix (Invitrogen; Thermo Fisher Scientific, Darmstadt, Germany) according to the manufacturer’s protocol. Quantitative reverse transcription PCR (RT-qPCR) was performed using TaqMan^®^ Gene Expression assays (Applied Biosystems; Thermo Fisher Scientific, Darmstadt, Germany) CK2α (CSNK2A1; Hs00751002_s1) and ribosomal RNA 18S (Hs99999901_s1 18S-FAM, Applied Biosystems, USA), which was used as the housekeeping gene. RT-qPCR reactions included 7.5 µL TaqMan Gene Expression Mastermix, 0.75 µL 2× TaqMan assay, RNase-free water, and 2 µL cDNA (100 ng) in a total volume of 15 µL. The thermocycling conditions were: 50 °C for 2 min, 95 °C for 10 min; this was followed by 50 cycles of 95 °C for 15 s and 60 °C for 1 min, in a StepOnePlus™ Real-Time PCR system (Applied Biosystems; Thermo Fisher Scientific, Inc.). StepOne version 2.0 software was used to measure the qPCR curves. Finally, Cq values were exported and analyzed in Microsoft Excel to determine the 2^−ΔΔCq^ values [28]. All experiments were performed at least three times.

### 2.14. Determination of Somatic Mutations

Somatic mutations in FFPE CRC tissue were analyzed by GenXPro (Frankfurt, Germany) using the commercially available TruSight One Expanded panel (Illumina) which is clinically applied to formalin fixed, paraffin-embedded derived DNA prior to diagnostic use and covers coding regions of 6794 genes and 16.6 Mb of genomic content [29]. The Illumina-Assay is validated for the detection of single nucleotide variants and small indels and allows the detection of mutations with variant allelic frequencies (VAF) as low as 5%. After isolation of genomic DNA from FFPE samples (see “Material and methods / DNA extraction”) GenXPro (Frankfurt, Germany) performed next-generation sequencing and bioinformatic data processing using the TruSight One Expanded panel (Illumina). Paired-end reads were mapped against the human reference genome GRCh38 (NCBI. “GRCh38—hg38—Genome—Assembly—NCBI”. ncbi.nlm.nih.gov. Retrieved 15 March 2019). Variant calling was performed using an in-house developed bioinformatics pipeline incorporating a Burrows-Wheeler Aligner (BWA) for alignment [30], as well as data pre-processing for variant discovery—GATK (broadinstitute.org)—somatic short variant discovery (SNVs + Indels)—GATK (broadinstitute.org)—and Annovar for variant annotation [31]. Somatic mutations were calculated as mutations per megabase (mut/mb).

### 2.15. Statistical Analysis

Unpaired two-tailed T-tests for normality, followed by Welch correction for uneven variations, and the Mann-Whitney test for non-normally distributed data were used to assess statistical significance. Survival data were plotted by the Kaplan-Meier method and were determined for statistical significance by the Log-rank-test.

All calculations were analyzed using the software GraphPad Prism 7 for Windows, Version 7.04 (GraphPad Software, La Jolla, CA, USA). The data shown are means ± SEM, unless otherwise stated, the following p-values were considered as statistically significant: * *p* < 0.05, ** *p* < 0.01, *** *p* < 0.001.

## 3. Results

### 3.1. The Majority of CRCs Showed Elevated CK2α Expression

Tissue from 143 MLH1-proficient and 22 MLH1-deficient CRCs and the available corresponding surrounding normal mucosa was analyzed for CK2α expression by immunohistochemistry. The CRCs showed three different pictures of CK2α expression (exemplarily shown in Figure 1): a significantly higher nuclear/cytoplasmic CK2α expression in the tumor tissue when compared to adjacent normal mucosa (Figure 1A,D), a significantly higher nuclear CK2α expression in the tumor tissue when compared to adjacent normal mucosa (Figure 1B,E), and a low nuclear/cytoplasmic CK2α expression in the CRC tissue, which was comparable to that of the corresponding normal tissue (Figure 1C,F).

All slides were scanned and the intensities of CK2α staining were quantified and analyzed using ImageJ as previously described [22]. Quantitation of CK2α was successful in 100% of cases (Appendix A). When comparing the CK2α levels in the 143 cases of our MLH1-proficient cohort (Appendix A) we found that 53.1% of MLH1-proficient CRCs express significantly higher nuclear/cytoplasmic CK2α level, 15.4% of the MLH1-proficient CRCs showed significantly high nuclear localized CK2α expression, while 31.5% of the MLH1-proficient cases showed low CK2α expression, similar to that of the normal adjacent mucosa (Figure 2A). By comparing the normal adjacent mucosa of these groups, we could also detect differences in CK2α levels. CK2α expression of tumor-surrounding normal mucosa of CRCs with high nuclear/cytoplasmic, as well as of those with high nuclear, CK2α expression showed a significant increase in CK2α expression when compared to the normal adjacent mucosa of CRCs with low nuclear/cytoplasmic CK2α expression (Figure 2B).

In the 22 cases of our MLH1-deficient group of CRCs, we detected the same three groups of differential CK2α-expressing CRCs (Appendix A, Appendix A). We found 54.5% of MLH1-deficient CRCs expressing significantly higher nuclear/cytoplasmic CK2α levels, 27.3% showing significantly high nuclear localized CK2α, and 18.2% harboring low nuclear/cytoplasmic CK2α expression when compared to the normal adjacent mucosa. Therefore, we concluded that MMR deficiency does not play a role either for the CK2α expression level nor for its localization.

Next, pairs of tumor and respective normal mucosa, which were available in 140 MLH1-proficient cases, were compared. As demonstrated in Figure 2C, the CK2α level of CRCs with high nuclear/cytoplasmic as well as high nuclear CK2α expression were significantly enhanced when compared to that of their corresponding normal adjacent mucosa. In contrast, CRCs with low nuclear/cytoplasmic CK2α expression showed similar CK2α levels when compared to the surrounding normal mucosa (Figure 2C).

We then separated the CRCs in different UICC stages and determined the intensity of CK2α in these UICC stages. Here, we detected that the overall intensities of high nuclear/cytoplasmic CK2α expression were similar in all UICC stages, but were significantly enhanced in UICC stage I, II, and III, as well as UICC stage IV. In contrast, high nuclear CK2α-expressing CRCs were completely absent in UICC stage I and were most abundant and significantly increased in UICC stage III and IV. Finally, the intensity of low nuclear/cytoplasmic CK2α expression was detectable in all UICC stages without differences (Figure 2D, Appendix A).

When we grouped the tumors according to the TNM system (T1–T4, Figure 2E, Appendix A) and compared the intensity level of CK2α in relation to T1–T4, we found that the differences between high nuclear/cytoplasmic or high nuclear and low nuclear/cytoplasmic CK2α intensities were significantly enhanced in all tumor stages, but high nuclear cases were absent in T1 and T2 (Figure 2E, Appendix A). The intensity of high nuclear/cytoplasmic CK2α in general was lowest in T1.

### 3.2. CK2α Quantity and Localization Allows Prediction of Overall Survival in Patients with CRC

In order to consider whether intratumoral CK2α protein levels might be related to overall survival in patients with CRC, survival outcomes of patients with high nuclear/cytoplasmic, high nuclear or low nuclear/cytoplasmic CK2α-expressing CRCs were separately analyzed by a Cox proportional hazards model. We included 140 patients with available survival data from the cohort of 143 patients with MLH1-proficient CRCs. The data were collected during aftercare appointments at the University Hospital Frankfurt. Data closure for the survival data was 30th April 2019. If patients did not attend their aftercare appointments at all or until that date at the University Hospital, survival data are missing or incomplete, respectively. In the considered 5-year survival outcome, only 50% of patients with CK2α high nuclear/cytoplasmic-expressing CRCs were alive after 42.9 months, and only 50% of patients with high nuclear CK2α-expressing CRCs were alive after 18.07 months (Figure 3A). In contrast, after 5 years, more than 60% of patients with low nuclear/cytoplasmic CK2α-expressing CRCs were still alive (Figure 3A). This trend of improved survival in the low nuclear/cytoplasmic CK2α-expressing group was maintained after adjusting for patient age and tumor stage in a multivariable proportional hazards model (Table 2). The overall survival of the patients with high nuclear/cytoplasmic CK2α expression when compared to those with low nuclear/cytoplasmic CK2α-expressing CRCs was significantly reduced (*p* = 0.0236), as well as the overall survival of patients with high nuclear CK2α expression when compared to patients with low nuclear/cytoplasmic CK2α-expressing CRCs (*p* = 0.0035). It therefore appears that low nuclear/cytoplasmic intratumoral CK2α protein levels may independently predict better overall survival in patients with CRC, while high nuclear intratumoral CK2α seems to be the worst predictor (Figure 3A, Table 2).

In all further studies, we then focused on a small exploratory panel of patients. We selected tumor and normal adjacent mucosa from patients a) from whom sufficient FFPE tissue was available and b) in whom the differences of CK2α expression were particularly clear: 11 cases which showed very high nuclear/cytoplasmic CK2α expression in the tumor tissues (patient 93; patient 103; patient 85; patient 43; patient 49; patient 50; patient 125; patient 132; patient 133; patient 139; patient 26; see Appendix A), 4 cases in which CK2α was highly expressed in the nuclei of the tumor tissues (patient 116; patient 98; patient 44; patient 89; see Appendix A), and 8 cases that showed very low nuclear/cytoplasmic CK2α expression in the tumor tissues (patient 162; patient 52; patient 135; patient 10; patient 23; patient 82; patient 148; patient 59; see Appendix A). These selected patients were first separately analyzed in terms of long-term survival. In total, long-term survival data were available from 21 of the 23 patients (Figure 3B, Table 2). Basically, a comparable picture to the whole cohort in terms of long-term survival was detectable. Again, patients whose tumors showed low nuclear/cytoplasmic CK2α expression survived significantly longer when compared to those with high nuclear/cytoplasmic or high nuclear CK2α expression (Figure 3B). In this cohort, only 50% of patients were alive after 18 months in the CK2α high nuclear/cytoplasmic-expressing group of patients, and only 50% of patients with high nuclear CK2α-expressing CRCs were alive after 10 months (Figure 3B). After 5 years, more than 60% of patients with low nuclear/cytoplasmic CK2α-expressing CRCs of our exploratory cohort were still alive (Figure 3B). The overall survival of patients with high nuclear/cytoplasmic CK2α-expressing CRCs was significantly decreased when compared to patients with low nuclear/cytoplasmic CK2α-expressing CRCs (*p* = 0.0438) (Figure 3B, Table 2). The trend of improved survival in the low nuclear/cytoplasmic CK2α-expressing group was again maintained after adjusting for patient age and tumor stage in a multivariable proportional hazards model (Table 2).

From the FFPE tissue of the selected 23 tumors and the corresponding normal adjacent mucosa, protein, DNA, and mRNA were isolated and examined in detail in the following analyses.

### 3.3. CK2α Overexpression in CRC Causes Enhanced Level of p-MLH1^S477^

As previously published, we were able to show in vitro that CK2α can phosphorylate MLH1 at amino acid position serine 477 and that this phosphorylation successfully inhibits MMR [9]. Here, we first used protein extracts of transiently pZW6(CK2α), pcDNA3.1+/MLH1 and pcDNA3.1+/PMS2 cotransfected HEK293T cells and verified by Western blotting and immunoprecipitation that CK2α overexpression can significantly increase the amount of p-MLH1^S477^ (Figure 4A and Appendix A). After fractionated cell extraction, we tried to define the cellular location where phosphorylated MLH1 is most dominantly produced and showed that p-MLH1^S477^ is mainly localized in the cytoplasm (Figure 4B and Appendix A). In contrast, in HEK293T cells, which expressed the unphosphorylatable MLH1^S477A^ variant, p-MLH1 was detectable neither in the cytoplasmic nor in the nuclear fraction (Figure 4B and Appendix A). The success of protein fractionation was verified by determination of Lamin β as nuclear protein control and γ Adaptin as a control for cytoplasmic proteins.

Next, the impact of CK2α on MLH1 phosphorylation was determined in vivo. After the successful isolation of protein extracts of good quality from FFPE tissue of the selected panel of our cohort, the detection of p-MLH1^S477^ was performed, whereby protein extracts of CRCs and normal adjacent mucosa were handled and analyzed in parallel (Figure 4C,D and Appendix A). p-MLH1^S477^ was detectable in all samples, but CRC tissues with high nuclear/cytoplasmic CK2α expression showed significantly higher amounts of p-MLH1 when compared to high nuclear or low nuclear/cytoplasmic CK2α-expressing tissue. High nuclear CK2α-expressing CRCs showed the lowest levels of p-MLH1 (Figure 4B,D). In contrast, all samples of normal adjacent mucosa showed a weak and comparable amount of p-MLH1^S477^ (Figure 4C,E). p-MLH1 levels of high nuclear/cytoplasmic, high nuclear, and low nuclear/cytoplasmic CK2α-expressing CRC tissue was correlated to normal adjacent mucosa and compared.

Due to the complexity of protein extraction from FFPE tissue and the fact that 2000 µg total protein was necessary for each immunoprecipitation, the analysis of p-MLH1^S477^ from FFPE tissue could only be performed one time, respectively.

### 3.4. High Nuclear/Cytoplasmic CK2α Expression Correlates with a Significant Increase of Somatic Mutation Rates

Since we could show that phosphorylation of MLH1 switches off MMR in vitro [9], we next tested in vivo if high CK2α expression—which we could verify to induce high p-MLH1-levels—is responsible for higher mutation rates in corresponding tumor tissue. To analyze this, we again used FFPE tissue of the selected 23 CRC patients and, in this case, isolated DNA. The resulting quality of DNA was too weak in two CRC samples, one of them with high nuclear/cytoplasmic and one of them with low nuclear/cytoplasmic CK2α expression. Therefore, tumor DNA of 21 patients was used and somatic mutations were determined using the TruSight One Expanded panel (Illumina) by the company GenXPro (Frankfurt, Germany). As shown in Figure 5A, somatic mutations were significantly increased in the CRCs with high nuclear/cytoplasmic CK2α expression when compared to low nuclear/cytoplasmic CK2α-expressing CRCs.

We further followed the question for the reason of differentially expressed CK2α levels. It has been shown that wildtype adenomatous polyposis coli (APC) protein is able to build a complex with CK2 and regulate CK2 activity and cell cycle progression possibly via CK2 stabilization, while mutated APC is not able to do this [32]. Therefore, we decided to focus separately on the mutation status of the APC gene that is included in the TruSight One Expanded panel (Figure 5B). Interestingly, all high nuclear/cytoplasmic CK2α-expressing CRCs of the selected panel of samples were mutated in APC, while the majority of CK2α nuclear/low cytoplasmic-expressing tumors were not mutated (Figure 5B, Table 3).

### 3.5. Changes in CK2α Protein Expression Partially Correlate with Varying CK2α mRNA Levels

To disclose the reason for the differential CK2α expression in CRCs, we further used RT-qPCR to analyze underlying mRNA levels. We again used the same selected high nuclear/cytoplasmic, high nuclear, and low nuclear/cytoplasmic CK2α-expressing tumor samples and corresponding normal adjacent mucosa from our patients’ cohort, as described above, and then isolated mRNA, generated cDNA, and determined the level of CK2α mRNA. The mRNA isolation from FFPE tissue of four patients was not successful, therefore, only mRNA of 19 patients could be used for RT-qPCR. The amount of CK2α mRNA was calculated by subtracting the mRNA level of normal adjacent mucosa from those of the tumor tissue, and were calculated as 2^−ΔΔCq^ values (Figure 6). As demonstrated in Figure 6, CK2α mRNA levels of high nuclear/cytoplasmic as well as high nuclear CK2α-expressing tumors tended to be higher than CK2α mRNA levels of low nuclear/cytoplasmic CK2α-expressing CRCs.

### 3.6. Somatic SNPs of the CK2α Promotor Region Detectable in CRC Cases

The CK2α mRNA results brought up the question if somatic mutations in the CK2α promoter might be the reason for differences in the CK2α mRNA level. Therefore, we again isolated genomic DNA from FFPE tissue of our panel of selected high nuclear/cytoplasmic, high nuclear, and low nuclear/cytoplasmic CK2α-expressing tumor samples, as well as normal mucosa, and amplified the CK2α promoter region -1850 to +364 (NC_000020.11:472498-543790; [26]) by PCR. Because of the weak quality of the FFPE DNA, we used ten pairs of primers (Table 2) for the amplification of ten overlapping fragments and determined the amplified DNA fragments by sequencing. Interestingly, we found several SNPs in the analyzed CRC promoter areas (Table 4). The positions of the SNPs are shown in Appendix A. Although several SNPs have been already detected in the CK2α core area (Appendix A), the detected single base pair alterations in our panel of patients are not described so far.

## 4. Discussion

The expression and activity of protein kinase CK2 is dramatically altered in numerous human tumor entities, making it an attractive candidate protein for targeted personalized molecular therapy. The large number of substrates of CK2 dictates that underlying molecular mechanisms are still largely not understood [33]. Recently, we have shown that phosphorylation of MLH1 at amino acid position serine 477 can be managed in vitro by CK2—precisely CK2α—and can switch off MMR [9]. In the present study, we performed in vivo CK2α analyses in a cohort of 165 CRC patients by correlating the CK2α expression, as well as transcription levels, with tumor mutation rates and survival time of patients, and determined the molecular mechanisms of differential intratumoral CK2α expression. In line with others, we found that a high percentage of CRCs showed significantly increased nuclear/cytoplasmic CK2α protein expression [34,35]. The association between CK2α overexpression and poor prognosis has also been described for numerous other tumors [36,37,38]. Interestingly, high nuclear/cytoplasmic CK2α expression could be matched in our study not only with poor survival in patients but also with significantly enhanced tumor mutation rates. Since a higher amount of p-MLH1 correlates with high nuclear/cytoplasmic CK2α, we hypothesize that cytoplasmic-localized CK2α phosphorylates MLH1, which causes its arrest in the cytoplasm and prevents its involvement and activity in MMR, and finally enhances the accumulation of mutations. Consistent with this hypothesis, we could show in vitro that CK2α overexpression correlates with enhanced p-MLH1 levels in the cytoplasm of human cell lines. The phenomenon that protein phosphorylation can promote cytoplasmic over nuclear localization has been described by many others before. Gao et al. e.g., showed that phosphorylation by Akt1 promotes cytoplasmic localization of the F-box protein of the E3 ubiquitin ligase complex Skp2 (S-phase kinase-associated protein 2) [39], Rodier et al. detected that p27 cytoplasmic localization is regulated by phosphorylation on Ser10 [40], and Xie et al. found that Protein kinase A (PKA) phosphorylation of Polypyrimidine tract-binding protein (PTB) at Ser-16 modulates the nucleo-cytoplasmic distribution of PTB [41]. Concerning the compartment in which MLH1 phosphorylation takes place, one might also assume that MLH1 is phosphorylated in the nucleus by nuclear CK2α in order to signal MLH1 for nuclear export. Enhanced selective removal of p-MLH1 from the nucleus would also explain a decreased MMR and, consequently, an increase in mutation rates. Nevertheless, since CRCs with enhanced high nuclear CK2α expression do not show increased levels of p-MLH1 and significantly less somatic tumor mutations when compared to high nuclear/cytoplasmic CK2α-expressing CRCs, this assumption seems rather unlikely. The exact mechanism, however, is not clear yet, and has to be analyzed in detail in future.

The picture of blockade of the MMR system in high nuclear/cytoplasmic CK2α-expressing CRCs is completed by the observation that all high nuclear/cytoplasmic CK2α-expressing CRCs of our small collective were also simultaneously mutated in the *APC* gene. Although mutations in the *APC* gene is one of the early changes in CRC progression, in the present case, a link to the loss of function of MLH1 might be drawn. Accordingly, Ahadova et al. analyzed the mutation signature in MMR-deficient CRCs of patients with Lynch syndrome and revealed that *APC* mutations commonly occur after loss of MMR [42]. Since APC has been described to be able to build a complex with CK2 and is capable of regulating CK2 activity [43], mutation-effected loss of regulatory APC / CK2 complex building might indirectly affect CK2α expression levels.

As discussed above, the ability of CK2α to phosphorylate and switch off MLH1 seems to be restricted to the cytoplasm, which might also give an explanation why CRCs, which dominantly expressed high nuclear intratumoral CK2α, are not able to phosphorylate MLH1, and did not show increased amounts of p-MLH1^S477^ or enhanced somatic mutation rates. However, why do these patients of our cohort nevertheless show the poorest survival time? Basically, the correlation of poor survival time of patients with high nuclear CK2α-expressing CRCs has been also described by Lin et al. [34], and Homma et al. recently demonstrated in breast cancer patients that clinicopathological malignancy was strongly correlated with elevated nuclear CK2α expression [44]. Concerning the underlying molecular mechanism, we speculate that the dominantly nuclear-overexpressed CK2α involves especially its function as a dynamic regulator for genes associated with cell cycle progression. Fitting with this, Schaefer et al. demonstrated a critical role of protein kinase CK2α in controlling DNA replication initiation and the expression levels of replicative DNA helicases [45], which suggests this having a dramatic impact on cellular processes, and can certainly contribute significantly to tumor progression. In addition, several studies demonstrated CK2α-dependent regulation of cancer stem cells. Tubi et al. showed that CK2α is critical for the sustenance of NF-κB, STAT3, and AKT/FOXO signaling pathways and leukemia stem cells’ survival, regulating the balance between apoptosis and cell cycle [46]. Zhang et al. detected that the inhibition of CK2α led to the reduction of a stem-like side population in human lung cancer cells [47]. Therefore, we assume that high nuclear CK2α expression might also have serious effects on the regulation of cancer stem cell markers.

The overall survival of patients with low nuclear/cytoplasmic CK2α-expressing CRCs was best in our cohort and this group also showed the lowest mutation rates when compared to high nuclear/cytoplasmic or high nuclear CK2α-expressing ones. Since we recently showed in vitro the correlation of a reduction of MMR by enhanced phosphorylation of MLH1 [9], the in vivo data demonstrated in the current study underline our speculation of CK2α-caused blocking of MLH1 function by phosphorylation at amino acid position serine 477. We therefore speculate that in between this subgroup of MSS CRCs, which generate enhanced tumor mutation rates caused by CK2α overexpression, are those patients who might benefit from PD-1 inhibitor treatment.

Investigating the reason for differential intratumoral CK2α expression, we detected that the transcription level of CK2α determined in the exploratory cohort of patients tended to be higher in CRCs, which already showed high nuclear/cytoplasmic or high nuclear CK2α protein expression level. In addition, we were able to identify in those tumors several somatic variants in the promoter region of CK2α. So far, only a few SNPs are described (https://www.genecards.org, accessed on 1 January 2022) and no specific CK2α promoter SNPs have been defined to be associated with modified promoter activity. If those SNPs we identified in the promoter region of CK2α of tumor tissue are responsible for the differential intratumoral expression of CK2α, this has to be investigated by using a larger cohort of CRCs in future. Of note, we identified a large amount of C/G to T/A changes in the analyzed CK2α promoter area, most frequently in cases of high nuclear/cytoplasmic CK2α-expressing CRCs. Fitting with this, Alexandrov et al. recently demonstrated a special MMR deficiency signature showing enhanced C-to-T transitions based on defects in the proofreading domain of DNA polymerase ε in CRCs and uterine tumors [48]. Moreover, several groups indicated that members of the apolipoprotein B mRNA catalytic subunit (APOBEC)/activation-induced deaminase (AID) family of DNA cytosine deaminases are a major source of C/G to T/A mutations [48,49,50]. Thus, one might hypothesize that a defective DNA polymerase ε or a deregulation of DNA deaminases is responsible for the observed enrichment of C/G to T/A changes in the promotor region of CK2α in the case of high nuclear/cytoplasmic CK2α-expressing CRCs. However, detailed molecular analyses are mandatory to proof underlying mechanisms.

Finally, a correlation of CK2α and co-morbidities in patients could not be analyzed in our cohort, since corresponding data were unfortunately not available. Future prospective studies, however, should include co-morbidities, especially viral infections such as SARS-CoV-2, which has been previously demonstrated to promote CK2 activation and cytoskeletal rearrangements [51].

In summary, we showed, for the first time, that intratumoral overexpression of CK2α is responsible for enhanced p-MLH1^S477^ levels, which seems to be causative for increased somatic tumor mutation rates. These data are starting points for further investigation to improve CRC therapy and patient survival.

## 5. Conclusions

In the present study, we showed that CK2α overexpression is a common phenomenon in CRCs, which is independent of the MMR status. In addition, we detected that high CK2α expression correlates with enhanced phosphorylation of MLH1 at amino acid position serine 477 and—potentially caused by CK2α promotor SNPs—this generates significantly more mutations in CRCs, presumably due to a reduction of the MMR mechanism, which is functional in low nuclear/cytoplasmic or high nuclear CK2α-expressing tumors. This reveals the importance of CK2α expression for the induction of somatic mutations and as a potential good additional diagnostic marker for an individualized therapy of patients with CRC.

## Figures and Tables

**Figure 1 cancers-14-01553-f001:**
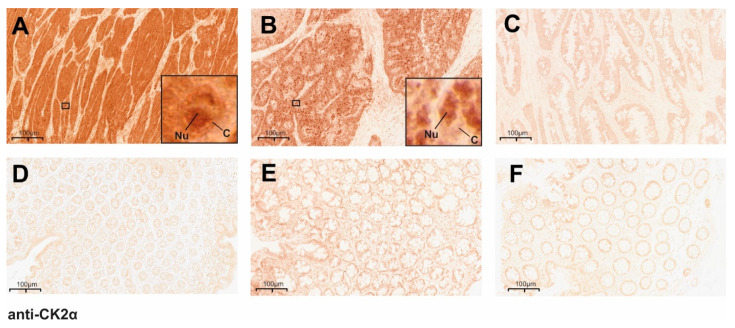
Immunohistochemical staining demonstrates significantly differential CK2α expression in CRCs. Paraffin-embedded, invasively growing CRCs and corresponding surrounding normal mucosa was analyzed for CK2α expression by immunohistochemical staining. Exemplarily shown are results of CK2α expression in (**A**–**C**) different CRC tissues and (**D**–**F**) the respective corresponding normal mucosa. Three different types of CK2α expression were detected: (**A**,**D**) CRC tissue shows very high nuclear/cytoplasmic CK2α expression and normal mucosa shows low CK2α level; (**B**,**E**) CRC tissue shows high nuclear CK2α expression and normal mucosa shows low/moderate CK2α level; (**C**,**F**) CRC tissue and normal mucosa both show comparable low nuclear/cytoplasmic CK2α expression. Original images were captured at 10-fold magnification, and the areas outlined by a quadrangle (**A**,**B**) show the magnification of individual cells for better visualization. C: cytoplasm; Nu: nucleus.

**Figure 2 cancers-14-01553-f002:**
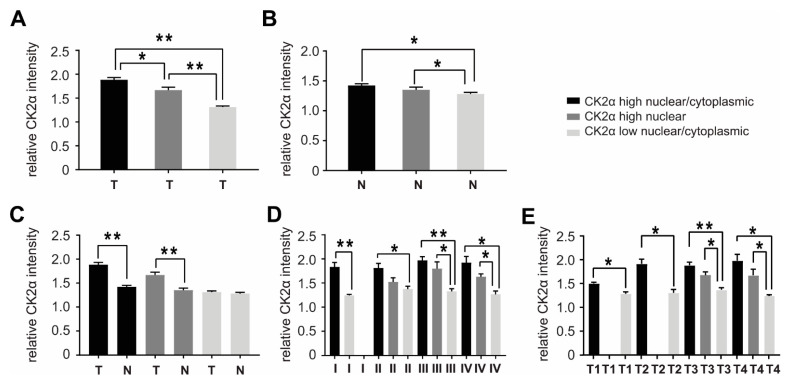
CK2α expression is significantly enhanced in a high proportion of CRCs. The intensity of CK2α expression in our MLH1-proficient cohort was determined by immunohistochemistry, analyzed using ImageJ and then compared in CRC tissue and corresponding normal adjacent mucosa. Tumors were grouped by (i) high nuclear/cytoplasmic, (ii) high nuclear or (iii) low nuclear/cytoplasmic CK2α-expressing CRCs. CK2α intensities were compared (**A**) only within all tumor tissues (T), (**B**) only within all normal mucosa (N), (**C**) between CRCs and the corresponding normal adjacent mucosa, respectively. Tumor tissues with high nuclear/cytoplasmic, high nuclear, or low nuclear/cytoplasmic CK2α expression were then divided and compared between different (**D**) UICC and (**E**) TNM stages (additional information see Appendix A). Statistical significance was assessed by GraphPad Prism 7 for Windows, Version 7.04 (GraphPad Software, La Jolla, CA, USA). The data shown are means ± SEM, *p*-values are two-sided and values <0.05 (*) or <0.01 (**) are considered statistically significant.

**Figure 3 cancers-14-01553-f003:**
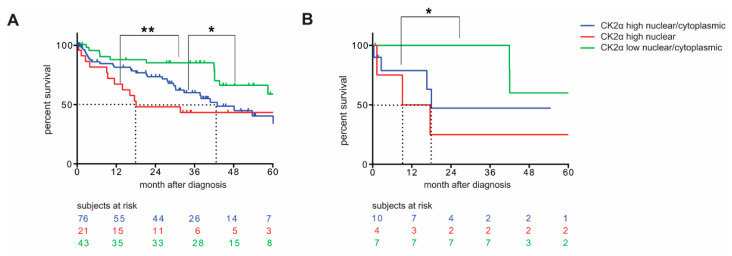
Level and localization of CK2α expression correlate with different overall survival in patients with CRC. CRC cases were grouped into high nuclear/cytoplasmic (blue line), high nuclear (red line) or low nuclear/cytoplasmic (green line) CK2α-expressing tumors and survival outcomes were analyzed by Cox proportional hazards model. (**A**) Comparison of survival outcomes within the whole CRC cohort; (**B**) Comparison of survival outcomes in a selected panel of patients, which were used for further exploratory analysis. Tumors of patients with the lowest survival outcome show high nuclear CK2α expression in the whole cohort, as well as in the selected panel of CRCs. Results with a *p*-value < 0.05 (*) or < 0.005 (**) are considered statistically significant.

**Figure 4 cancers-14-01553-f004:**
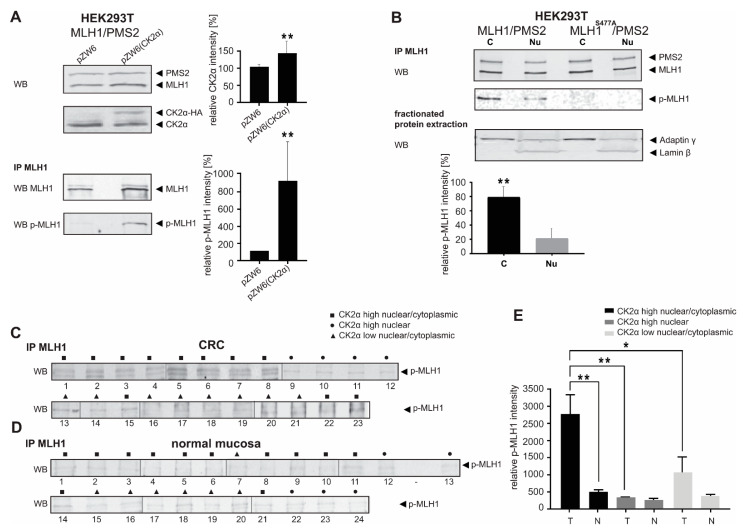
Enhanced p-MLH1^S477^ levels in CRC are associated with high nuclear/cytoplasmic CK2α expression. Protein extracts were isolated from (**A**) transiently pcDNA3.1+/MLH1, pcDNA3.1+/PMS2 and pZW6(CK2α), or empty pZW6 cotransfected HEK293T cells, (**B**) cytoplasmic and nuclear fractions of transiently pcDNA3.1+/MLH1 and pcDNA3.1+/PMS2 or pcDNA3.1+/MLH1^S477A^ and pcDNA3.1+/PMS2 cotransfected HEK293T cells, (**C**) FFPE CRC tissue, and (**D**) FFPE normal adjacent mucosa of high nuclear/cytoplasmic, high nuclear and low nuclear/cytoplasmic CK2α-expressing cases. After immunoprecipitation with anti-MLH1 antibody, p-MLH1 was detected by Western blotting. p-MLH1 levels were quantified (mean ± S.D.) using the Multi Gauge V3.2 program. (**E**) Comparison of p-MLH1 levels of FFPE CRC and normal mucosa of high nuclear/cytoplasmic, high nuclear and low nuclear/cytoplasmic CK2α-expressing cases, respectively. Statistical significance was assessed by GraphPad Prism 7 for Windows, Version 7.04 (GraphPad Software, La Jolla, CA, USA). *p*-values are two-sided and values <0.05 (*) or <0.001 (**) are considered statistically significant; (**C**) 1: patient 93; 2: patient 103; 3:patient 85; 4: patient 43; 5: patient 49; 6: patient 50; 7:patient 125; 8: patient 132; 9: patient 89; 10: patient 44; 11: patient 98; 12: patient 116; 13: patient 59; 14: patient 148; 15: patient 26; 16: patient 82; 17: patient 23; 18: patient 10; 19: patient 135; 20: patient 52; 21: patient 162; 22: patient 139; 23: patient 133; (see Appendix A). (**D**) 1: patient 93; 2: patient 103; 3: patient 85; 4: patient 125; 5: patient 50; 6: patient 43; 7: patient 23; 8: patient 139; 9: patient 133; 10: patient 132; 11: patient 101; 12: patient 95; 13: patient 98; 14: patient 141; 15: patient 82; 16: patient 52; 17: patient 10; 18: patient 135; 19: patient 59; 20: patient 148; 21: patient 26; 22: patient 116; 23: patient 44; 24: patient 89; (see Appendix A). T: Tumor tissue; N: normal adjacent mucosa; C: cytoplasm; Nu: nucleus.

**Figure 5 cancers-14-01553-f005:**
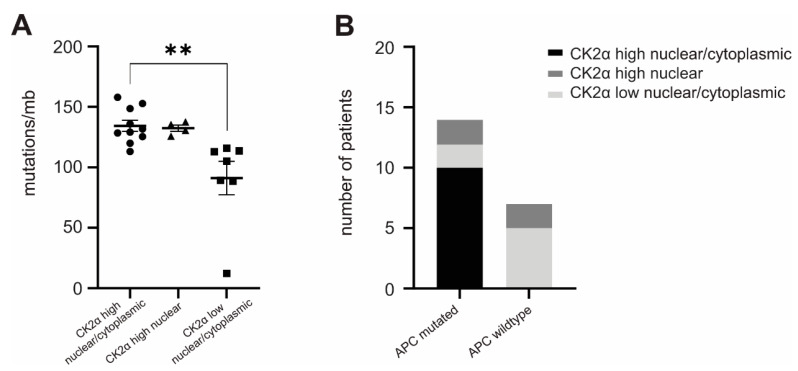
Impact of differential CK2α expression on tumor mutation rates. DNA isolated from FFPE tissue of 21 CRCs was analyzed for somatic mutations using the TruSight One Expanded panel (Illumina); (**A**) Mutation rates of high nuclear/cytoplasmic or high nuclear CK2α-expressing tumors were compared to low nuclear/cytoplasmic CK2α-expressing CRCs. The amount of mutations was significantly enhanced in high nuclear/cytoplasmic CK2α expressing CRCs; (**B**) The panel of analyzed genes includes the APC gene. The mutation status in the respective three differentially expressing CK2α CRCs is shown here. All high nuclear/cytoplasmic expressing CK2α CRCs show mutations in the APC gene. Statistical significance was assessed by GraphPad Prism 7 for Windows, Version 7.04 (GraphPad Software, La Jolla, CA, USA). *p*-values are two-sided and values <0.001 (**) are considered statistically significant.

**Figure 6 cancers-14-01553-f006:**
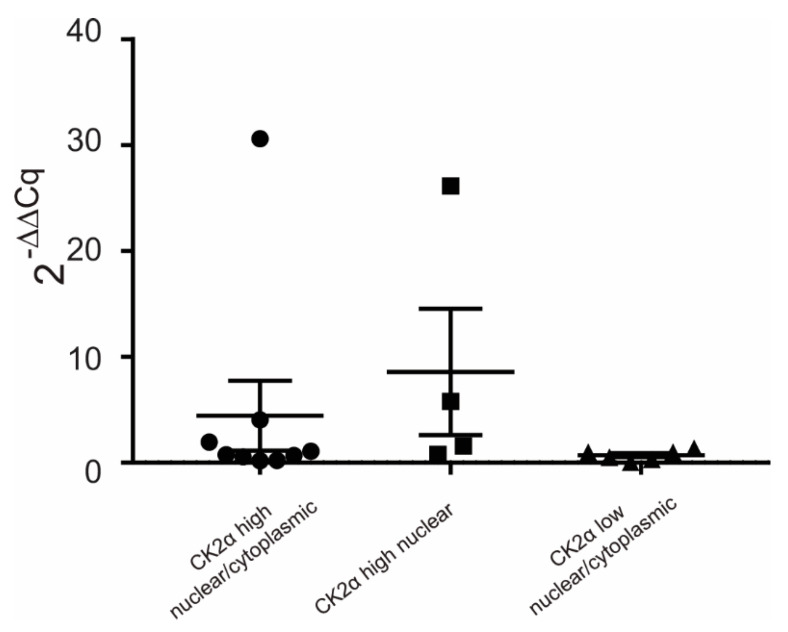
Analysis of CK2α mRNA expression in CRCs. mRNA was isolated from FFPE CRC tissue and corresponding normal adjacent mucosa and transcribed into cDNA. RT-qPCR was performed using TaqMan^®^ Gene Expression assays for CK2α (CSNK2A1; Hs00751002_s1) and ribosomal RNA 18S (Hs99999901_s1 18S-FAM, Applied Biosystems, USA) as housekeeping gene. The relative CK2α mRNA levels tended to be higher in high nuclear/cytoplasmic and in high nuclear CK2α-expressing CRC tissue. Low nuclear/cytoplasmic CK2α-expressing CRCs show the weakest CK2α mRNA expression. Data are expressed as means ± SEM as appropriate.

**Table 1 cancers-14-01553-t001:** Primers used for CK2α promoter amplification.

Localization—CK2α Gene Promoter Region [26]	Forward Primer	Reverse Primer
-1850 to -1555	5′-AGCACTTATTGCTACCTGAA-3′	5′-AATCCCAAAGTTTCTGGGAAGC-3′
-1629 to -1334	5′-CCAAAAAGATACGTTCGAGAGG-3′	5′-AAAGAGGCACCTCTTCCCCA-3′
-1404 to -1105	5′-CCCTGAGGCCATCACTATAA-3′	5′-TGATAAAAGCTGAAGCGTCTAA-3′
-1179 to -888	5′-CACCTCTGTCCCACCAGAGGTG-3′	5′-CTTCTCTTACTGTCACCTCA-3′
-962 to -657	5′-TAGAGGAAAGGATCCCTGAA-3′	5′-CTCATCATGGTCTCCCTATGGT -3′
-734 to -446	5′-CAAGTGAAGAGTTTGGGCTATC -3′	5′-CCTAGGAAGGGCATGGCGCA-3′
-520 to -202	5′-GGAAGGAATTGGGCCTTGGT-3′	5′-ACGAACCTCCCATTAGGTGAAC-3′
-282 to -4	5′-CAGCTGGGTGAAGTGTGGGAAA-3′	5′-AGACAGCTTCCGACTCCGCC-3′
-78 to +221	5′-CTAAGGTTACAATAGGACA-3′	5′-TATCCTGGGCCCACCCCACCCG-3′
+155 to +364	5′-GCTTCCACCACAGGTACCTAGG-3′	5′-CCGCCCTGAGGGGTGGCCCC-3′

**Table 2 cancers-14-01553-t002:** Overall survival in the cohort analyzed by multivariable Cox regression.

**Overall Survival in the Whole Cohort**		
**Variables**	**Hazard Ratio** **[95% CI]**	**Significance Level** **[*p*-Value]**
CK2α protein (high nuclear/cytoplasmic vs. low nuclear/cytoplasmic)	2.033 [1.114–3.71]	0.0236
CK2α protein (high nuclear vs. low nuclear/cytoplasmic)	3.178 [1.286–7.854]	0.0035
CK2α protein (high nuclear/cytoplasmic vs. high nuclear)	0.6943 [0.3365–1.433]	0.1147
**Overall Survival in the Selected Cohort**		
**Variables**	**Hazard Ratio** **[95% CI]**	**Significance Level** **[*p*-Value]**
CK2α protein (high nuclear/cytoplasmic vs. low nuclear/cytoplasmic)	3.564 [0.6278–20.24]	0.0873
CK2α protein (high nuclear vs. low nuclear/cytoplasmic)	5.694 [0.7611–42.6]	0.0438
CK2α protein (high nuclear/cytoplasmic vs. high nuclear)	0.5019 [0.09712–2.694]	0.4469

**Table 3 cancers-14-01553-t003:** APC mutation status.

	High Nuclear/ Cytoplasmic	High Nuclear	Low Nuclear/ Cytoplasmic
APC mutated	10	2	2
APC wildtype		5	2

**Table 4 cancers-14-01553-t004:** Detected somatic SNPs in CRC located in the CK2α promoter.

Patient Number	CK2α Expression	Genomic SNP Localization *	SNP Position Correlated to Transcription Initiation Site ***
103	high nuclear/cytoplasmic	g.4315G>A g.4333G>A g.4453C>T	-686 -668 -548
43	high nuclear/cytoplasmic	g.3406T>C **g.3551C>T **** g.3557C>T/C g.3563A>G g.3585C>T g.3631C>T	-1595 **-1450** -1444 -1438 -1416 -1370
125	high nuclear/cytoplasmic	g.3396G>A/G g.3418G>A g.3561C>T/C g.3625C>T g.4742 C>T g.4842C>T g.4892C>T g.4939A>G	-1609 -1583 -1440 -1376 -259 -159 -109 -62
133	high nuclear/cytoplasmic	g.4792A>G	-209
139	high nuclear/cytoplasmic	g.3405G>A g.3425 G>A g.3533G>A g.3945G>A g.4018G>A g.4078G>A g.4343A>G g.4371G>A	-1596 -1576 -1468 -1056 -983 -923 -658 -630
98	high nuclear	g.4498G>A	-503
44	high nuclear	g.3430C>T **g.3551C>T **** g.3591C>T g.3661C>T g.3944G>T	-1571 **-1450** -1410 -1340 -1057
52	low nuclear/cytoplasmic	g.3424G>A g.3510G>A g.4814C>T	-1577 -1491 -187
135	low nuclear/cytoplasmic	g.4805C>T	-196
10	low nuclear/cytoplasmic	g.4449G>A g.4463G>A **g.5154C>T ****	-552 -538 **+154**
23	low nuclear/cytoplasmic	**g.5154C>T ****	**+154**
82	low nuclear/cytoplasmic	g.4819G>A	-182
59	low nuclear/cytoplasmic	g.3524G>A	-1477

* reference sequence NC_000020.11:472498-543790. ** bold marked SNPs were detected in different patients. *** [26].

## Data Availability

Data supporting already reported CK2α promotor SNPs are available under https://www.genecards.org, accessed on 1 January 2022.

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
