# Peer review of "High Expression of Casein Kinase 2 Alpha Is Responsible for Enhanced Phosphorylation of DNA Mismatch Repair Protein MLH1 and Increased Tumor Mutation Rates in Colorectal Cancer"

_cancers, 2022, doi:10.3390/cancers14061553_

Round 1

Reviewer 1 Report

Colorectal cancer (CRC) is associated with DNA mismatch repair (MMR) deficiency. The serine/threonine casein kinase 2 alpha (CK2α) can phosphorylate and inhibit MMR protein MLH1 in vitro. This study aimed to analyze the relevance of CK2α for MLH1 phosphorylation in vivo. Around 50% of CRCs were identified to express significantly increased nuclear/cyto- plasmic CK2α. High nuclear/cytoplasmic CK2α level could be significantly correlated with reduced 5-year survival outcome of patients, increased MLH1 phosphorylation and enriched somatic tumor mutation rates. Overall, this study demonstrated in vivo that enhanced CK2α leads to an increase of MLH1 phosphorylation, higher tumor mutation rates and is an unfavorable prognosis for patients.However there are some questions that needs to be addressed

  • Please include sclae bars for Figure 1
  • Do the authors found any correlation between the CK2 alpha expression and other co-morbidities in the patients?
  • Whether the medications taken by the patients will influence the levels/expression of CK2 alpha?
  • How would the levels of CK2 alpha will influence the stem cell markers expression such as LGR5, DCLK1?

Author Response

Responses to Reviewer #1

Overall, this study demonstrated in vivo that enhanced CK2α leads to an increase of MLH1 phosphorylation, higher tumor mutation rates and is an unfavorable prognosis for patients.

We thank the reviewer for acknowledging the contribution regarding the role of CK2a concerning enhanced phosphorylation of MLH1 and the association to increased tumor mutation rates and worse prognosis of patients.

  • Please include scale bars for Figure 1

We modified Figure 1 and included scale bars. This new Figure 1 is shown in the revised version of this manuscript.

  • Do the authors found any correlation between the CK2 alpha expression and other co-morbidities in the patients?

No, a correlation between CK2 alpha expression and other co-morbidities could not be analyzed, since information besides tumor data was not collected for this cohort. Current and future prospective studies, however, should include any co-morbidities especially viral infections like SARS-CoV-2, which was shown to promote CK2 activation and cytoskeletal rearrangements (Bouhaddou M, Memon D, Meyer B, et al. The Global Phosphorylation Landscape of SARS-CoV-2 Infection. Cell. 2020;182(3):685-712)

To emphasize on this, we added in the discussion section of revised version:

Finally, a correlation of CK2α and co-morbidities in patients could not be analyzed in our cohort since corresponding data were unfortunately not available. Future prospective studies, however, should include co-morbidities especially viral infections like SARS-CoV-2, which has been previously demonstrated to promote CK2 activation and cytoskeletal rearrangements [51].

  • Whether the medications taken by the patients will influence the levels/expression of CK2 alpha?

In the manuscript we mentioned (2. Materials and Methods, 2.1 Patients): “All patients underwent colonic resection with curative intent, and without prior exposure to neoadjuvant therapies.”

In order to make this statement more clear we now enlarged this paragraph in the revised version:

All patients underwent colonic resection with curative intent. Individuals with prior exposure to neoadjuvant chemotherapy were excluded from the study, in order to avoid interference from cytoreductive therapies that may conceivably alter tumor genetics.

  • How would the levels of CK2 alpha will influence the stem cell markers expression such as LGR5, DCLK1?

The expression of cancer stem cell proteins is of great importance for the generation and progression of tumors. However, neither LGR5 nor SCLK1 have been described as substrates for CK2α so far. Whether an increased amount and activity of CK2α – especially of nuclear localized CK2α - has any influence on the expression of these two proteins or responsibly regulatory pathways cannot be concluded yet and should be investigated in future studies.

Since in general there are many studies, which demonstrate that CK2α has significant influence on several stem cell marker. We therefore added the following paragraph in the discussion section:

In addition, several studies demonstrated CK2α-dependent regulation of cancer stem cells. Tubi et al. showed that CK2α is critical for the sustenance of NF-κB, STAT3 and AKT/FOXO signaling pathways and leukemia stem cells survival, regulating the balance between apoptosis and cell cycle [46]. Zhang et al. detected that the inhibition of CK2α led to the reduction of a stem-like side population in human lung cancer cells [47]. Therefore, we assume that high nuclear CK2α expression might also have serious effects on the regulation of cancer stem cell markers.

Reviewer 2 Report

In this report the authors link CK2 phosphorylation of MLH with mutation activity in colorectal cancers. They build on a previous study. The findings are intriguing and appropriate for dissemination in this journal. However, this reviewer believes that certain clarifications are warranted before publication.

For Fig.1 please label one panel with N for nucleus and C for cytoplasm. It is not clear to this reviewer where the nucleus is in these cells especially in panels A and B. The authors are concluding that nuclear to cytoplasmic expression levels are important so then it is important for the reader to see where the nucleus and/or cytoplasm is in this figure.

More information should be given on how survival was monitored (section 3.2). Were all patients from which expression data was captured monitored, were there missing data, etc. Also, this sentence (line 387): “Median overall survival was 42.9month in the CK2α high nuclear/cytoplasmic expressing group of patients, 18.07 month in patients with high nuclear CK2α expressing CRCs and was not reached in the low nuclear/cytoplasmic CK2α expressing group (Figure 3A).” What was not reached? Somewhat awkward, please rephrase. This reviewer is further confused by apparent contradictory conclusions. For example, at line 419 it says, “The overall survival of patients with high nuclear/cytoplasmic CK2α ex-pressing CRCs was significantly decreased compared to patients with low nuclear/cytoplasmic CK2α expressing CRCs (p=0.0438)”. This sentence appears to draw the opposite conclusion than the one at line 387: 42.9 months is the highest survival given. Please explain.

Another clarification is needed on the status of MLH phosphorylation. The authors asset that phosphorylation arrests the protein in the cytoplasm (line 576). What is not clear is whether phosphorylation is a signal for nuclear exit. For example, if phosphorylation happens in the nucleus, does the protein migrate to the cytoplasm? That would explain why there is an increase in mutation rate: no available repair factor left in the nucleus.

I am struck by the overwhelming majority of C/G>T/A transition type mutation shown in Table 4. C>T transitions have been proposed to represent clocklike mutational signatures in human cancers (PubMed ID 26551669). However, the mutations in Table 4 are not transcription associated (e.g. strand specific) because they are in a non-transcribed area. Thus, C>T can be G>A within one round of replication and vice versa. DNA polymerase errors can produce C>T transitions (PubMed ID 17121822). The authors should expand on this finding, even if speculating in the discussion on its significance.

Author Response

 Responses to Reviewer #2

The findings are intriguing and appropriate for dissemination in this journal.

We thank the reviewer for this very positive assessment of our manuscript and the following constructive comments.

1) For Fig.1 please label one panel with N for nucleus and C for cytoplasm. It is not clear to this reviewer where the nucleus is in these cells especially in panels A and B. The authors are concluding that nuclear to cytoplasmic expression levels are important so then it is important for the reader to see where the nucleus and/or cytoplasm is in this figure.

We improved Figure 1 to clarify nuclear and cytoplasmic CK2α expression. We zoomed out areas and marked with “Nu” nucleic and “C” cytoplasmic parts of tumor cells. 

2) More information should be given on how survival was monitored (section 3.2). Were all patients from which expression data was captured monitored, were there missing data, etc.

The following text is added in the revised version:

We included 140 patients with available survival data from the cohort of 143 patients with MLH1 proficient CRCs. The data were collected during aftercare appointments at the University Hospital Frankfurt. Data closure for the survival data was April 30th, 2019. If patients did not attend their aftercare appointments at all or until that date at the University Hospital, survival data were missing or incomplete, respectively.

3) Also, this sentence (line 387): “Median overall survival was 42.9 month in the CK2α high nuclear/cytoplasmic expressing group of patients, 18.07 month in patients with high nuclear CK2α expressing CRCs and was not reached in the low nuclear/cytoplasmic CK2α expressing group (Figure 3A).” What was not reached? Somewhat awkward, please rephrase.

To clarify this, we changed this sentence to:

In the considered 5-year survival outcome, only 50% of patients with CK2α high nuclear/cytoplasmic expressing CRCs were alive after 42.9 month and only 50% of patients with high nuclear CK2α expressing CRCs after 18.07 month (Figure 3A). In contrast, after 5 years more than 60% of patients with low nuclear/cytoplasmic CK2α expressing CRCs were still alive (Figure 3A).

We also modified the text describing the 5-year survival outcome of our exploratory cohort.

In this cohort, only 50% of patients were alive after 18 month in the CK2α high nuclear/cytoplasmic expressing group of patients, and only 50% of patients with high nuclear CK2α expressing CRCs after 10 month (Figure 3B). After 5 years more than 60% of patients with low nuclear/cytoplasmic CK2α expressing CRCs of our exploratory cohort were still alive (Figure 3B).

4) This reviewer is further confused by apparent contradictory conclusions. “The overall survival of patients with high nuclear/cytoplasmic CK2α ex-pressing CRCs was significantly decreased compared to patients with low nuclear/cytoplasmic CK2α expressing CRCs (p=0.0438)”. This sentence appears to draw the opposite conclusion than the one at line 387: 42.9 months is the highest survival given. Please explain.

We agree that the statement made in this sentence (similar like this of line 387) might be misleading and useful information was missing earlier in the text. As we now changed the statement of sentence (line 387), we think the correlation should be clear here.

5) Another clarification is needed on the status of MLH1 phosphorylation. The authors asset that phosphorylation arrests the protein in the cytoplasm (line 576). What is not clear is whether phosphorylation is a signal for nuclear exit. For example, if phosphorylation happens in the nucleus, does the protein migrate to the cytoplasm? That would explain why there is an increase in mutation rate: no available repair factor left in the nucleus.

We agree with the reviewer that it is not clear, if MLH1 is phosphorylated in the nucleus or in the cytoplasm and if phosphorylation of MLH1 is a signal for nuclear exit or avoids the import of MLH1.

To discuss this point, we enlarged the discussion section and included:

Concerning the compartment in which MLH1 phosphorylation takes place one might also assume that MLH1 is phosphorylated in the nucleus by nuclear CK2α in order to signal MLH1 for nuclear export. Enhanced selective removal of p-MLH1 from the nucleus would also explain a decreased MMR and consequently an increase of mutation rates. Nevertheless, since CRCs with enhanced high nuclear CK2α expression do not show increased level of p-MLH1 and significantly less somatic tumor mutations compared to high nuclear/cytoplasmic CK2α expressing CRCs this assumption seems rather unlikely. The exact mechanism, however, is not clear yet and has to be analyzed in detail in future.”

6)            I am struck by the overwhelming majority of C/G>T/A transition type mutation shown in Table 4. C>T transitions have been proposed to represent clocklike mutational signatures in human cancers (PubMed ID 26551669). However, the mutations in Table 4 are not transcription associated (e.g. strand specific) because they are in a non-transcribed area. Thus, C>T can be G>A within one round of replication and vice versa. DNA polymerase errors can produce C>T transitions (PubMed ID 17121822). The authors should expand on this finding, even if speculating in the discussion on its significance.

We thank the reviewer for this comment and for pointing out the literary references. A discussion about the cause of the type of changes identified in the SNPs seems very appropriate to us.

We have therefore added the following section to the discussion:

Of note, we identified a large amount of C/G to T/A changes in the analyzed CK2α promoter area, most frequently in cases of high nuclear/cytoplasmic CK2α expressing CRCs. Fitting with this, Alexandrov et al. recently demonstrated a special MMR deficiency signature showing enhanced C-to-T transitions based on defects in the proofreading domain of DNA polymerase ϵ in CRCs and uterine tumors [48]. Moreover, several groups indicated that members of the apolipoprotein B mRNA catalytic subunit (APOBEC)/activation-induced deaminase (AID) family of DNA cytosine deaminases are a major source of C/G to T/A mutations [48-50].Thus, one might hypothesize that a defective DNA polymerase ϵ or a deregulation of DNA deaminases are responsible for the observed enrichment of C/G to T/A changes in the promotor region of CK2α in case of high nuclear/cytoplasmic CK2α expressing CRCs. However, detailed molecular analyses are mandatory to proof underlying mechanisms.”

Round 2

Reviewer 2 Report

The authors have made significant changes to this version. This reviewer is satisfied.